# Influence of Atmospheric Conditions on Mechanical Properties of Polyamide with Different Content of Recycled Material in Selective Laser Sintering

**DOI:** 10.3390/polym14122355

**Published:** 2022-06-10

**Authors:** Ana Pilipović, Petar Ilinčić, Ante Bakić, Janoš Kodvanj

**Affiliations:** 1Faculty of Mechanical Engineering and Naval Architecture, University of Zagreb, 10000 Zagreb, Croatia; petar.ilincic@fsb.hr (P.I.); janos.kodvanj@fsb.hr (J.K.); 2Probotica d.o.o., 10000 Zagreb, Croatia; ante.bakic@probotica.hr

**Keywords:** ageing, atmospheric conditions, flexural properties, orientation, polyamide, recycling, selective laser sintering, tensile properties, weathering conditions, water absorption

## Abstract

The price of material is an important factor when selecting the additive polymer procedure. In selective laser sintering (SLS), the price can be reduced by the recycling of material, i.e., with different shares of original and recycled material, as well as by the orientation of the product during manufacturing. Numerous tests warn that orientation in the direction of z axis should be as low as possible to reduce the total price of the product. The product also has to satisfy the influence of atmospheric conditions to which it is exposed during its lifetime, i.e., UV radiation and humid environment. UV light, with sun being its most common source, and average humidity in different parts of the world can be approximately from 20% to 90%, depending on time, day and geographic location. In this work, the test specimens have been made of original, mixed and 100% recycled material and then exposed to the influences of UV radiation and water absorption. After having been exposed to atmospheric conditions for a longer time, the mechanical properties of the polyamide products made by selective laser sintering were tested. The results show that exposure to UV radiation reduces tensile elongation at all ratios of recycled material and orientation of 70–90% except in the z direction, while in flexural deformation it is the other way around. The effect of water was observed only between the 7th–14th day of absorption with a decrease in strength until the deformation did not change.

## 1. Introduction

Additive manufacturing allows the production of parts with very complex shapes, the making of which had been limited until these procedures were developed. Additive procedures are being intensively developed daily. Here, the number of available materials is limited, and their properties differ a lot from the properties of the materials of finished products. Therefore, it is necessary to know the mechanical properties of the product material [1]. In selecting the processing procedures, it is necessary to take into consideration four criteria: the desired material, size and number of products, time of manufacturing and cost of production [2].

In selective laser sintering (SLS) of polymer products, the price of material is not the most important factor, but it does play a significant role in the overall analysis, i.e., the final price of the product, and therefore the recycling of material is becoming an increasingly important factor, as well as the ratio of the shares in combining the original and the already used material. Since, in the SLS procedure, the powder is used as support structure, this unused powder can be reused. It is possible to mix this material in the ratio of 67% of used and 33% of original powder in polyamide, [3] but there is no mention of the possibility of using only 100% used material that had already passed through several cycles, and how it would affect the product properties. Dotchev et al. [4] concluded that in the case of using recycled powder, which is exposed to higher temperature and longer time, experiences a much higher deterioration rate. The temperature and the time to which the unsintered material was exposed are the most influential parameters for the powder aging, but this refers to aging during the product manufacturing process itself—exposing the material to the temperature required in the processing chamber for polyamide. According to manufacturer *EOS GmbH* powder has to be sifted before mixing in order to remove undesired impurities and possible coarse-grained accumulations of powder. [5] Many authors recommend the use of original powder only, since the addition of the already used material results in poorer surface quality. In literature [4,6], it has been determined that surface roughness is related to the type of material (original or recycled) and melt flow rate (MFR), and an accepsuppurface in PA 12 material is achieved at an MFR of 18 g/10 min.

The most frequently used material in the SLS procedure is polyamide (PA). Polyamide absorbs humidity, which affects the mechanical properties and product swelling. Humidity absorption is related to the lowering of the glass transition temperature *T*_g_. The hygroscopic nature of polyamide has been explained through the polyamide structure, which is mainly crystalline with amorphous areas. The polar amide group (-NHCO-) forms crystalline areas; however, not all electrons are uniformly distributed among atoms, which leads to the attraction of the charged areas and linking of polymer chains. However, water is also a polar molecule and when polyamide absorbs water, the molecules of water are mixed with the areas in the polymer chain and poorly connect with the C=O and H–N links. This reduces polarity between chains causing an increase in chain mobility, which leads to the reduction of strength and increased flexibility. Absorbed water, therefore, acts as a softener. As a result of the increase in chain mobility, the glass transition temperature *T*_g_ is reduced, also causing the reduction of amorphous areas around *T*_g_ and in this way increasing the crystalline areas. (e.g., polyamide 6 (PA 6) stored in dry conditions has *T*_g_ about 60 °C, whereas when stored in wet conditions, *T*_g_ is reduced to room temperature or lower). The quantity of water absorption depends on the concentration of the amide group in the molecular chains. In the usual SLS procedures, PA 11 or PA 12 is used (maximal water absorption of product manufactured by classical manufacturing procedures is 0.7 to 0.8% [7]) which absorb less humidity than other polyamides, such as PA 6 (maximal absorption is 9.5–10.5% [7]) [8]. Most of the literature related to water/humidity absorption considered it for PA 6, and only a minor part for PA 12. However, the papers [8,9] explain that although water absorption causes a reduction in the strength of PA 6, it increases the tensile strength of PA 12. However, test specimens have been tested after only 24 h, which is not the case in practice, so that it is necessary to define what the mechanical properties are after a long period of water influence.

The higher the degree of crystallinity, i.e., with the increase in the degree of order in the structure there is less water absorption and a lower influence of humidity on the polymer properties. The more polar groups present in the polymer matrix, the greater will be its absorption affinity. Water absorption causes softening, especially of the amorphous part of the polyamide [10].

Ageing is a change, depending on time, in amorphous polymers or an amorphous phase in crystalline polymers as result of the mobility of polymer chains. When polymers are normally cooled from a temperature higher than *T*_g_ to a temperature lower than *T*_g_, the tendency is to achieve the actual volume (i.e., dimension). However, if the polymer is cooled rapidly, as in the case of laser sintering, it cannot shrink fast enough to retain the volume balance and therefore, the procedure must be continued slowly until balance is achieved [8]. At the same time the mobility of the polymer chains is drastically reduced in comparison to raised temperature, but the molecules still have low mobility and thus reach slowly the balanced state. The usual ageing effect is the increase in density, module of elasticity and reduction of ductility, strain at break and creep. Ageing in amorphous polymers occurs below *T*_g_, and in crystalline ones, above *T*_g_ of the amorphous phase [8]. In laser sintering, melting and crystallization can occur during the procedure, so that eventually further cooling is necessary.

In the SLS process, there is also an essential sintering window, melting temperature and enthalpy which can be analyzed from differential scanning calorimetry (DSC) curves. Uniform shrinkage and proper crystallization can be obtained by keeping the powder bed at a high temperature for some time even after sintering. Polymer powder depicts slight overlapping peaks during the heating and cooling phase of the DSC plot if operated at a slower crystallization rate. In cases in which the build temperature is too close to the onset of crystallization, the parts have larger temperature gradients and show warpage and parts will distort after release from the hot bed. In cases in which the build temperature is too close to the onset of melting, the hot melt acts as hot spots. The sintering of the surrounding solid powder occurs on the surfaces, resulting in lateral growth of the parts [11,12]. For semi-crystalline polymers usually used in SLS processing, this implies that crystallization (*T*_c_) should be inhibited during processing as long as possible, at least for several sintered layers [13].

A lot of work has concentrated on the impact of exposure to ageing (i.e., temperature) during production, such as the work of Wudy, K. and Drummer, D. [14] where it was concluded that the used PA 12 material in SLS can lead to unwanted effects on the produced component properties, such as uneven surfaces so called “orange peel” surfaces. To reduce or avoid changes to the PA 12 material during SLS, a responsible ageing mechanism should be understood. Higher build temperature and longer cumulative build time increase molecular weight distribution, where the influence of build time is more significant.

There is little work on the mechanical properties, especially flexural properties after the products have been made and applied for a long time because of the very nature of polymeric materials, and especially polyamide. In this paper, the mechanical properties of the product after it has been exposed to atmospheric conditions such as UV radiation and water absorption, i.e., moisture, are analyzed.

## 2. Materials and Methods

The tensile properties are determined according to HRN EN ISO 527: 2012 standard. The standard defines the equations to calculate the tensile properties and the dimensions of the test specimen. Flexural properties are tested according to standard HRN EN ISO 178:2011 and three-point testing has been applied. Test specimens, shown in Figure 1, were produced with the SLS printer *EOS Formiga P100*. For production of test specimens three different ratios of material PA 2200 from *EOS GmbH* (Krailling, Germany) were used:-100% of original polyamide 12;-mixed material (marked with M);-100% recycled material after third cycle of usage (marked with R).

**Figure 1 polymers-14-02355-f001:**
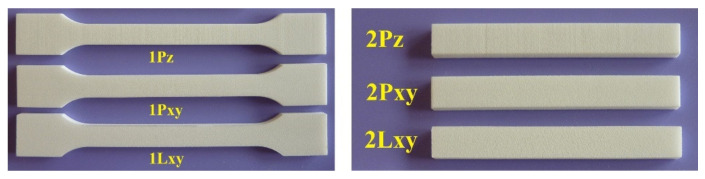
Test specimen manufactured by SLS process.

In the mixed material, original and recycled material in ratio 50:50% were mixed in an industrial mixer at room temperature. Duration of mixing process was 60 min.

To include the influence of layer orientation, test specimens of each material were printed in three different layer orientations in the working area (Figure 2):-Lxy–test specimen in plane xy with the height in z direction 4 mm;-Pxy–test specimen in plane xy with the height in z direction 10 mm;-Pz–test specimen in direction of z axis with the height 150 mm.

**Figure 2 polymers-14-02355-f002:**
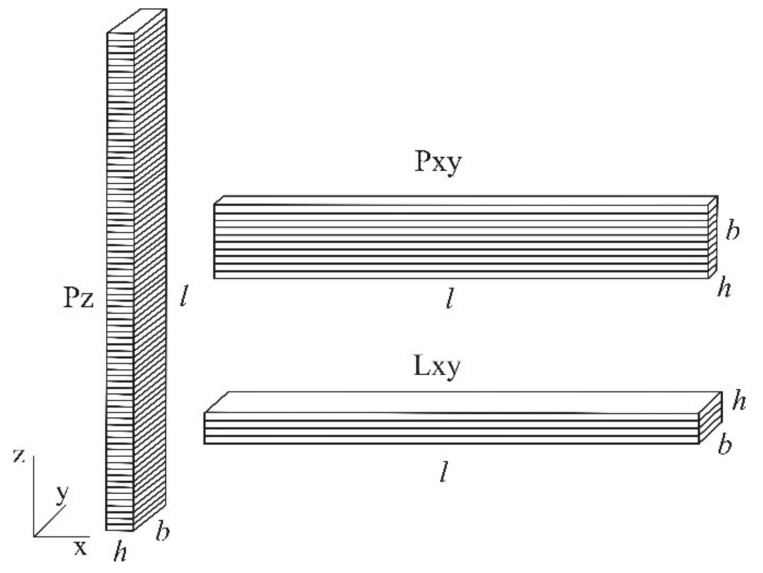
Layer orientation.

Printing parameters were equal for all materials:-chamber temperature 172 °C;-layer thickness 0.1 mm;-alternating layer scan direction;-beam offset 0.15 mm;-energy density *ED* = 0.0564 J/mm^2^;-laser power *P* = 21 W;-laser beam speed *v* = 2500 mm/s;-hatch distance *h* = 0.25 mm.

Mechanical properties were determined with the testing machine Messphysik Beta 50–5 of maximum loading force 50 kN, equipped with the control unit EDC 100. All tests were performed at a room temperature of 23 °C. To determine the tensile properties, the test specimen was gripped in the jaws of the tensile-testing machine (Figure 3a) and extended by force *F*. Strain was measured by video-extensometer. For testing of flexural properties (Figure 3b) the test specimen needed to be supported on two supports and loaded with force *F* in the middle, until the test specimen broke or until the deflection reached the agreed value *S*_c_. Testing speed for tensile and flexural properties was *v* = 5 mm/min.

Many polymer materials have to be protected against environmental conditions, e.g., sources of heat, oxygen, water and, especially, UV light. Although many polymers do not absorb directly ultraviolet radiation, the product surface contains some components that can absorb UV light and thus stimulate oxidation degradation of polymers. The most common source of UV radiation is the sun which causes a reduction in durability and undesired changes in the properties of the material. Artificial light can also affect the properties. UV radiation can cause fracture of chains in the material structure. This chemical process is called photo-oxidation and consists of two different processes: photolysis—which includes absorption of UV light and shaping of free radicals during the fracture of molecular links [15].

Exposure to light sources in the laboratory was performed according to HRN EN ISO 4892:2004 standard on the device SOLARBOX 1500e, manufactured by Erichsen, without wetting of specimens (Figure 4a), I radiation 550 W/m^2^ with wavelength from 300 nm to 800 nm, chamber temperature 65 °C.

Test specimens for tensile properties were placed into the chamber so that, after 1000 h spent in the chamber, they could be used for testing of mechanical properties and determining the impact of accelerated ageing on the properties.

In Europe, according to ASTM D3-424 standard, 1 h of laboratory testing in a UV chamber corresponds to 24 h of natural exposure to UV light. Testing performed after 1000 h in the chamber corresponds to 1000 days of natural exposure, i.e., 2.8 years. For the sake of comparison, field tests have been performed for the duration of 120 days (Figure 4b).

For all mechanical tests, three test specimens were made and the mean value and standard deviation were calculated. The results of all tests for all three test specimens and the calculated mean values and standard deviations can be found in the Appendix A.

## 3. Results

### 3.1. Tensile Properties of the Original Material

After longer exposure of test specimens to UV light at 100% of original polyamide, the tensile properties are reduced (Figure 5) in all three orientations. It is interesting that in orientation Pz the tensile strain at break remained the same after 500 h and 1000 h spent in the chamber, and after natural ageing, whereas in orientations Lxy and Pxy it was reduced by 70–90%. That is, with longer exposure to UV light the strain in orientations Lxy and Pxy was reduced.

From Figure 5, one can further conclude that test specimens exposed to natural weathering conditions (rain, wind, UV radiation) keep their good properties. Tensile strength was reduced only by 10%, whereas strain and tensile modulus in orientation Lxy and Pxy were reduced by almost half the value in comparison to the situation before exposure to weathering conditions.

In practice, many products are exposed to rain, i.e., humid atmosphere. With water absorption after 28 days at room temperature in all three orientations, the tensile and ultimate strength were reduced, tensile strain at break rose, but only in orientations Pxy from 28 to 35%. For comparison, testing was performed with test specimens that before tearing had been kept at a controlled temperature in a furnace at 100 °C for 2 h. Such test specimens showed a reverse situation—tensile and ultimate strength increased, but the tensile strain at break was reduced, in orientations Lxy and Pxy, whereas in orientation Pz the tensile strain at break remained the same (Figure 6).

However, when comparing water absorption in the period of 1, 4, 7, 14 and 28 days, it is noticeable that in the period of 14 days, the test specimens kept the same properties, and the reduction of stress was noted only after 28 days (Figure 7).

### 3.2. Tensile Properties of the Mixed Material

In different weathering conditions (humid atmosphere, ageing), the mixed material kept approximately the same values of stress, whereas strain was reduced by about 50%. Tensile strength in orientation Pz was lower than 42 MPa, while in the other two orientations (Lxy and Pxy) it was about 48 MPa. Strain depends significantly on the product orientation, since in Pz the strain is only 5% compared to orientation Lxy in which the strain is 25% (Figure 8). The tensile modulus in different orientations remained approximately the same, but it was affected by weathering conditions, where ageing reduced the initial tensile modulus from 1.8 GPa to 1.2 GPa.

### 3.3. Tensile Properties of 100% Recycled Material

The tensile strength was the lowest after exposure to 100% humid atmosphere (i.e., water absorption after 28 days); however, in orientations Lxy and Pxy, water increased the tensile strain at break. The values of the mechanical properties in the laboratory and natural ageing did not differ significantly (Figure 9). The tensile modulus under the influence of different weathering conditions was reduced from the initial 1.7 GPa to 1.1 GPa and behaved the same as in the case of the mixed material.

### 3.4. Flexural Properties of the Original Material

With the exposure to UV light, all the flexural properties increased (strength, strain and flexural modulus). After only 500 h under UV light no significant changes in product orientations Lxy and Pxy were noted, whereas in orientation Pz flexural strain at break fell by 32%, which could be caused by orientation, the smaller layer area and significantly higher number of layers compared to Lxy and Pxy orientation. After a longer exposure of 1000 h to UV light in orientations Lxy and Pxy, the flexural strength, flexural strength at break, flexural strain at break and flexural modulus rose, whereas in orientation Pz flexural strain at break continued to fall, but the strengths rose (Figure 10).

In the humid atmosphere, the flexural properties were significantly reduced, particularly flexural strength and flexural modulus. It should be noted that flexural strain at break remained the same (Figure 11). By curing at 100 °C for approximately 6 h the flexural properties, i.e., flexural strength was increased, depending on the product orientation, by 1 MPa to 5 MPa.

However, if a gradual reduction of flexural properties in humid atmosphere was observed, per days (1, 4, 7, 14 and 28 days) one could notice that from the 1st day to the 7th day, the flexural strength was insignificantly reduced, and only after that (Figure 12, period of 14 days and 28 days) one could notice a major fall in strength and flexural modulus. However, unlike the strength and the module, the flexural strain at break remained the same during the entire period of keeping the product in 100% humid atmosphere (Figure 12).

### 3.5. Flexural Properties of the Mixed Material

Figure 13 shows the diagram of flexural stress–strain for the mixed material. The diagram shows that, same as in the original material, the worst properties were after 28 days of water absorption by the product in 100% humid atmosphere in all three orientations. Flexural strain at break remained the same both before exposure to atmospheric conditions and after UV radiation and water absorption.

### 3.6. Flexural Properties of 100% Recycled Material

Figure 14 shows flexural properties in 100% recycled material for all three orientations. The figure shows that UV radiation, either in chamber or naturally, raised the flexural strength (flexural strength and flexural strength at break) and flexural modulus *E*_f_, and flexural strain at break remained the same. However, during water absorption, the strength started to decrease by 15–16% depending on the product orientation.

### 3.7. DSC Analysis

For a better understanding of the sintering window, DSC analysis was made. Specimens were analyzed with the device *Mettler Toledo DSC 823e*. Every test sample with the mass of approx. 10 mg was heated with the heating rate of 10 °C/min in two cycles and then cooled down in a temperature range from 0 to 210 °C and N_2_ atmosphere.

From the thermograms obtained in the first and second heating cycles, the values of melting temperature (*T*_m_) and melting enthalpy (*ΔH*_m_) were determined, while from the thermograms obtained in the cooling cycle crystallization temperature (*T*_c_) and associated crystallization enthalpy (*ΔH*_c_) were determined. The data obtained in the second heating cycle are taken as relevant considering that the data obtained in the first heating cycle was to overcome the thermal history of sample preparation. In Figure 15, Figure 16 and Figure 17, thermograms of the tested samples are shown and results of the *T*_m_, *ΔH*_m_, *T*_c_ and *ΔH*_c_ are given in Table 1.

From the thermograms of the first heating cycle, it could be concluded that there were nonsignificant differences in the shape of the thermal transition which correspond to the melting of the crystalline phase of the polyamide. The values of melting enthalpy indicated the amount of crystal domains in the polyamides. Higher values of melting enthalpy, i.e., a higher crystal structure, were found in the mixed material (50:50 original and recycled polyamide).

After the second heating cycle, the values of melting temperature were similar for all tested samples. Melting enthalpy for materials before exposure and after water absorption indicated the same polyamide structure. Enthalpy before exposure to atmospheric conditions and after water absorption behaved logically: the highest in the original material, then mixed and the lowest in 100% recycled material. However, it is interesting to note that recycled material had the highest enthalpy value after UV ageing. 

During the cooling cycle, the exothermic transition of crystallization occurs in a wide temperature range. The crystallization temperatures were similar in all tested samples (*T*_c_ ≅ 140 °C). Similarly, crystallinity values, *T*_c_, indicated the same rate of crystallization. A little higher value of the crystallization enthalpy points to a slightly higher crystal phase in the material before exposure and after water absorption (*ΔH*_c_ ≅ 57 J/g) while lower values of crystallization enthalpy of material after UV ageing indicate a slightly lower crystal phase (*ΔH*_c_ ≅ 49–53 J/g). In the cooling phase, it was also observed that the recycled material had the highest values of crystallization enthalpy after UV ageing.

## 4. Discussion

In the SLS procedure, it was found that the printing strategy is important, and the best mechanical properties had products oriented at an angle of 0° [16,17].

During the selective laser sintering, a considerable amount of powder residue is generated. To optimize the production costs, this residue should be reused, but the properties of the product should remain the same as with the original material. Authors Wang L. et al. [18] tried to determine whether this powder could be reused with the retention of the mechanical properties. In their research, they investigated the processing of SLS powder residue into filaments for extrusion-based additive manufacturing. To maintain or increase the mechanical properties, they added milled carbon fiber (mCF) to recycled polyamide in a quantity of 30 wt.%.

In the work of the author Feng L. et al. [19] to reduce the percentage of polyamide (PA 12) that remains unused in the SLS process, they made a filament out of that recycled material and applied it in the fused deposition modeling (FDM) process. Their conclusion was that in comparison with the fresh PA 12, the mechanical properties of the recycled PA 12 powder were only slightly reduced, but the melt flow rate (MFR) decreased by 77%.

According to Dooher et al. [20] over period of 6 months, changes in color (discoloration) appeared, which was also observed here after ageing.

The impact of weathering conditions on material properties is shown in Figure 18 for tensile strength *R*_m_ and tensile modulus *E*, where an average value for each material was calculated from the values obtained for each orientation. Tensile strength and tensile modulus were highest when using only the original material, whereas the values of mixed and recycled material did not differ. According to Czelusniak et al. [21], with the optimal processing parameters when applying 50:50% mixed polyamide, a tensile strength of 45 MPa could be obtained, which roughly corresponds to these results. By applying the original material and drying, the strength can be increased to 50 MPa and more.

It is interesting to note that in the mixed material, the tensile strength increased after a longer time of UV radiation, and the tensile modulus, in the case of water absorption.

Original material: with longer exposure to UV light in the case of the original material, the tensile properties were reduced in all three orientations. In orientation Pz the tensile strain at break remained the same after 500 h and 1000 h spent in the chamber, and after natural ageing, whereas in orientations Lxy and Pxy it was reduced by 70–90%. That is, with the longer exposure to UV light in orientations Lxy and Pxy, the material strain was reduced.

The product kept the same tensile properties in humid atmosphere up to 28 days. After 28 days of water absorption at room temperature in all three orientations, the tensile and ultimate strength were reduced, but the tensile strain at break increased, only in orientations Pxy from 28 to 35%. However, by curing the product at 100 °C over a time of 2 h, the tensile and ultimate strength rose, but the tensile strain at break decreased, in orientations Lxy and Pxy, whereas in orientation Pz the tensile strain at break remained the same.

Mixed material: the mixed material in different weathering conditions (humid atmosphere, UV radiation) kept approximately the same strength, whereas the strain was reduced by about 50%. Tensile strength in the case of orientation Pz was lower than 42 MPa, while in the other two orientations (Lxy and Pxy) it was about 48 MPa. The strain also depended a lot on the orientation of the product, since in Pz orientation the strain was only 5%, in comparison to Lxy in which the strain was 25%. The tensile modulus in various orientations remained approximately the same, but with ageing, the initial tensile modulus fell from 1.8 GPa to 1.2 GPa.

According to the paper by Zárybnická et al. [22], where the authors wanted to determine how the orientation in the working space of the machine affects the mechanical properties of the mixed material (50:50), a tensile strength of 44 MPa for the orientation Lxy was obtained, which was 4 MPa lower than in our tests, while the values for orientation in the z direction (Pz) were identical. It is interesting to note that the authors obtained a tensile modulus value of only 1300 MPa, which was almost 30% lower compared to our test. Even with the recycled material, higher values were obtained than in the work of Zárybnická et al. [22].

The results of a study by Panda et al. [23] showed the results of tensile strength at a break of 25–33 MPa depending on the orientation, which is a deviation from the values obtained in this paper by 25–43% (tensile strength values are 3644-MPa).

Recycled material: tensile strength was the lowest after exposure to 100% humid atmosphere (i.e., water absorption after 28 days). In orientation Lxy and Pxy, the water raised the tensile strain at break due to the increased orientation of the macromolecules. Due to weathering conditions, the tensile modulus was reduced from the initial 1.7 GPa to 1.1 GPa, and it behaved in the same way as in the mixed material.

Influence on mechanical properties when reusing the powder leftover from the process chamber up to four times was investigated in the study of authors du Maire, P. et al. [24]. Tensile test specimens were produced and tested. The results showed a decrease in the ultimate tensile strength with repeated reuse of the leftover powder. After four times of the powder reuse, the strength increased again, which is not the case here. The authors could not explain such material behavior and will continue to investigate such a material behavior. The investigations revealed a high mechanical anisotropy as the strength was highly dependent on the orientation of the component in the process chamber.

By comparing different shares of original material, one can notice that the recycled material had the lowest flexural strength *σ*_fm_ and flexural module *E*_f_, while the highest values were in the material mixed in ratio 50:50 of original and recycled material in different atmospheric conditions (Figure 19). Flexural strength and flexural module in Figure 19 were given as an average value for each material, calculated from the values obtained for each orientation.

Original material: exposure to UV light resulted in an increase of all flexural properties (stress, strain, and flexural modulus), whereas after water absorption of 28 days, they were reduced, particularly the flexural strength and flexural modulus. It is interesting to note that the flexural strain at break remained the same. By curing at 100 °C for approximately 6 h, flexural strength was increased by up to 5 MPa, depending on the product orientation.

Only after seven days of water absorption was a more significant drop in flexural strength and flexural modulus observed. However, unlike strength and modulus, flexural strain at break remained the same during the entire period of keeping the product in 100% humid atmosphere, which could lead to the conclusion that with a higher percentage of water absorption flexural modulus and flexural strength decrease, most probably due to the softening effect caused by water molecules on the material.

Mixed material: in the mixed material as well as in the original material, the properties were lowest after 28 days of water absorption in 100% humid atmosphere in all three orientations. Flexural strain at break remained the same both before exposure to atmospheric conditions and after UV radiation and water absorption.

Recycled material: with UV radiation, in the chamber or naturally, in the recycled material the flexural strength and flexural strength at break and flexural modulus *E*_f_ decreased, and flexural strain at break remained the same. Likewise, during water absorption the strength started to fall by approximately 16%, depending on the product orientation.

Authors Dooher et al. [20] concluded that material becomes more brittle during ageing, as the elongation at break and toughness shows a decrease which corresponds to this analysis, but only in the case of tensile strain at break, while the flexural deformation does not change significantly. When it comes to the orientation of the product in the z axis (Pz), such a conclusion is different, but this orientation itself has significantly lower values of tensile strains even before exposure to atmospheric conditions.

## 5. Conclusions

The type of material as well as orientation have a big influence on the tensile properties. Mixed and recycled materials have in the case of UV radiation almost equal tensile strength and tensile modulus. However, in the case of water absorption, the recycled material has the worst properties, since it absorbs the largest amount of water. In subsequent tests, the water absorption tests in the recycled material should be repeated and the absorption percentage should be checked. Before exposure to weathering conditions, either UV light or water absorption, the products made only of original material have the best tensile properties, but after a certain time of exposure to weathering conditions there comes an improvement of some properties in the mixed and recycled material. This is likely to be assigned to the so-called healing effect which is usual in the initial phases of the action of physically active media. The penetration of the molecules of absorbed water, first of all into amorphous areas, enables abandonment of internal stresses resulting, e.g., from the procedures of material preparation and partial arrangement of macromolecules in micro-areas, which leads to the improvement of the properties [25].

Regarding the product orientation it may be concluded that the strengths do not differ excessively, but that the tensile strain at break in orientation Pz is of low values compared to the other two orientations, only 5%. This occurs because during tension the test specimen in orientation Pz breaks exactly between two layers.

The highest tensile strength *R*_m_ of 51 MPa is achieved by curing the products after manufacture, minimally for 2 h at 100 °C.

By comparing different shares of original material, it can be noticed that the recycled material has the lowest flexural strength *σ*_fm_ f and flexural modulus Ef, while mixed material features the highest values in 50:50 ratio of original and recycled material in different atmospheric conditions.

Additional strengthening by UV radiation increases the flexural strength since polymer molecules are interconnected into chains.

With UV radiation the properties rise. It is interesting that such behavior is featured also by the worst orientation Pz. The highest flexural strength of 74 MPa was achieved by exposure to UV radiation for 1000 h.

From the performed tests related to the type of material, one can conclude that the properties are reduced with the use of recycled material only, but they still remain within satisfactory limits, so that it is recommended not to throw away the material that has been used several times, but rather to use it for some less demanding products.

## Figures and Tables

**Figure 3 polymers-14-02355-f003:**
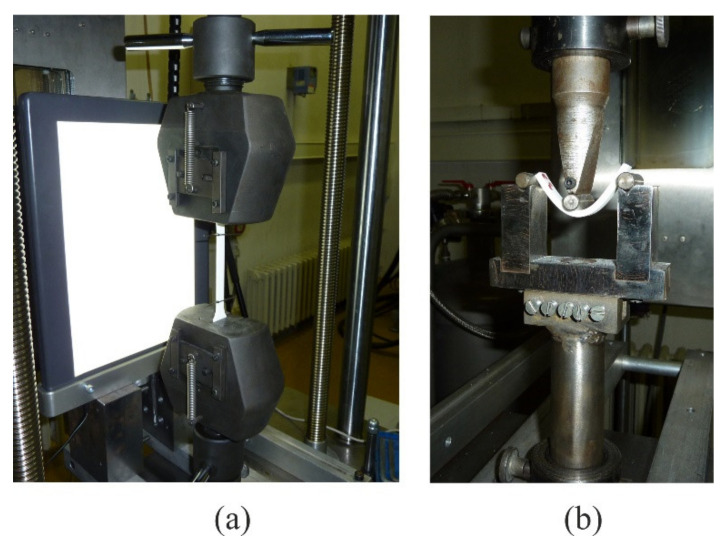
Jaws for tensile and flexural testing: (**a**) jaws for tensile testing, (**b**) supports for flexural testing.

**Figure 4 polymers-14-02355-f004:**
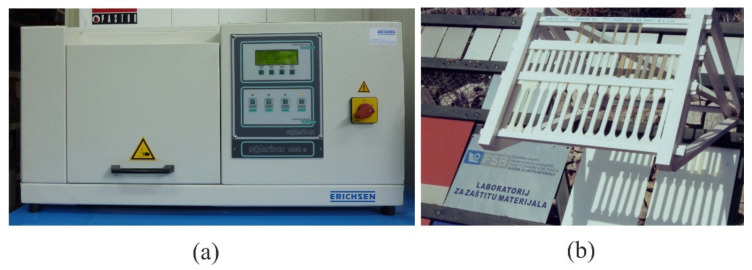
Chamber for testing of exposure to sources of light and field tests for the influence of atmospheric conditions: (**a**) chamber for laboratory exposure to weathering condition, (**b**) natural exposure to weather condition, location island Murter, Croatia.

**Figure 5 polymers-14-02355-f005:**
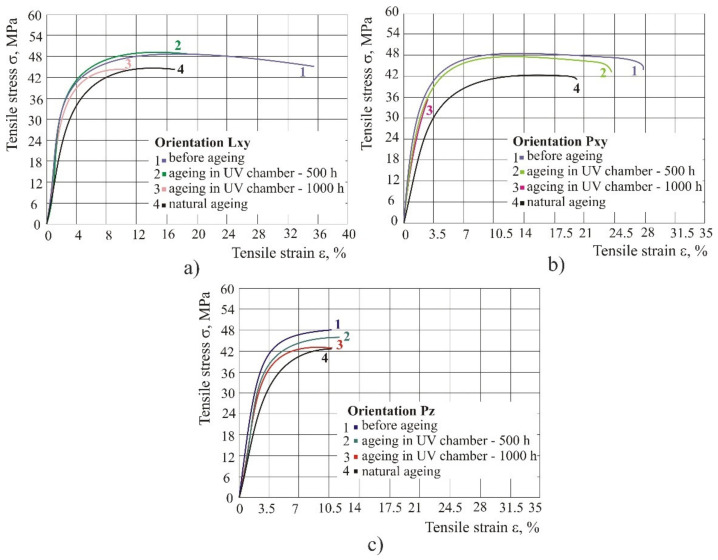
The diagram of tensile stress–strain of test specimens with the use of original material in case of UV radiation: (**a**) orientation Lxy, (**b**) orientation Pxy, (**c**) orientation Pz.

**Figure 6 polymers-14-02355-f006:**
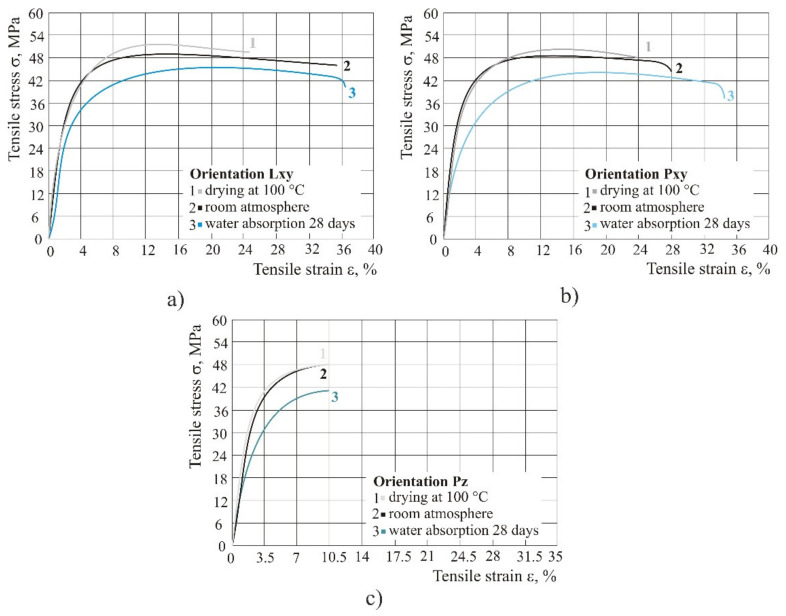
Comparison of different weathering conditions (dry, normal and 100% humid atmosphere): (**a**) orientation Lxy, (**b**) orientation Pxy, (**c**) orientation Pz.

**Figure 7 polymers-14-02355-f007:**
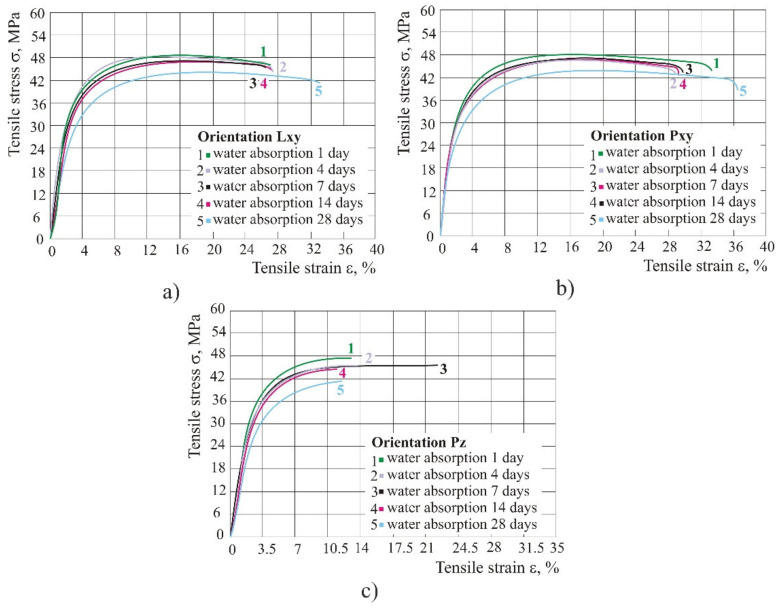
Diagram of tensile stress–strain in the case of water absorption: (**a**) orientation Lxy, (**b**) orientation Pxy, (**c**) orientation Pz.

**Figure 8 polymers-14-02355-f008:**
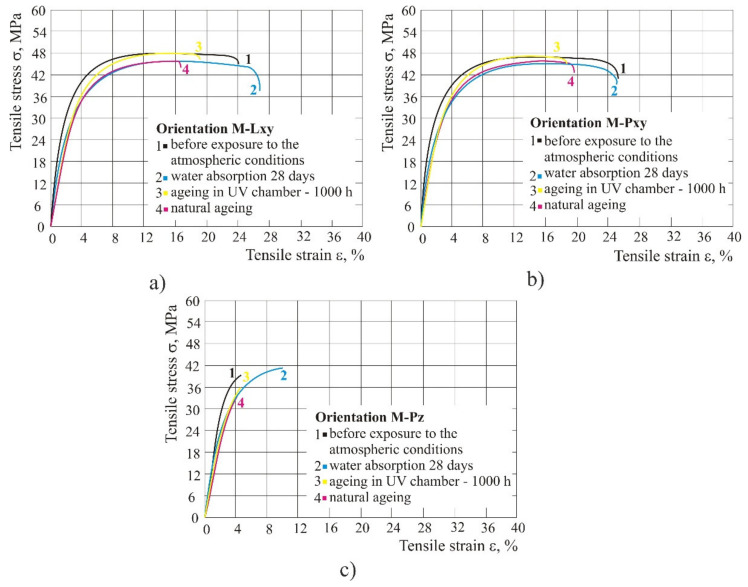
Impact of different weathering conditions on mixed material: (**a**) orientation Lxy, (**b**) orientation Pxy, (**c**) orientation Pz.

**Figure 9 polymers-14-02355-f009:**
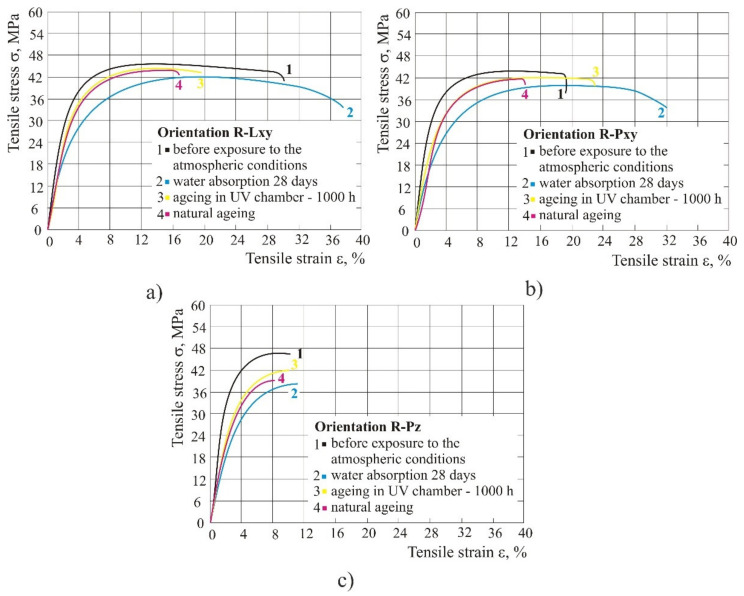
Impact of different weathering conditions on recycled material: (**a**) orientation Lxy, (**b**) orientation Pxy, (**c**) orientation Pz.

**Figure 10 polymers-14-02355-f010:**
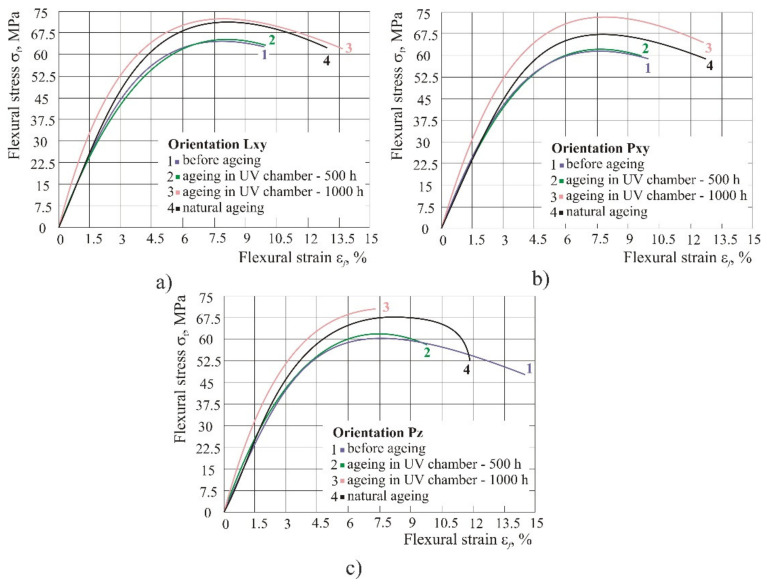
The diagram of flexural stress–strain of test specimens with the use of original material in the case of UV radiation: (**a**) orientation Lxy, (**b**) orientation Pxy, (**c**) orientation Pz.

**Figure 11 polymers-14-02355-f011:**
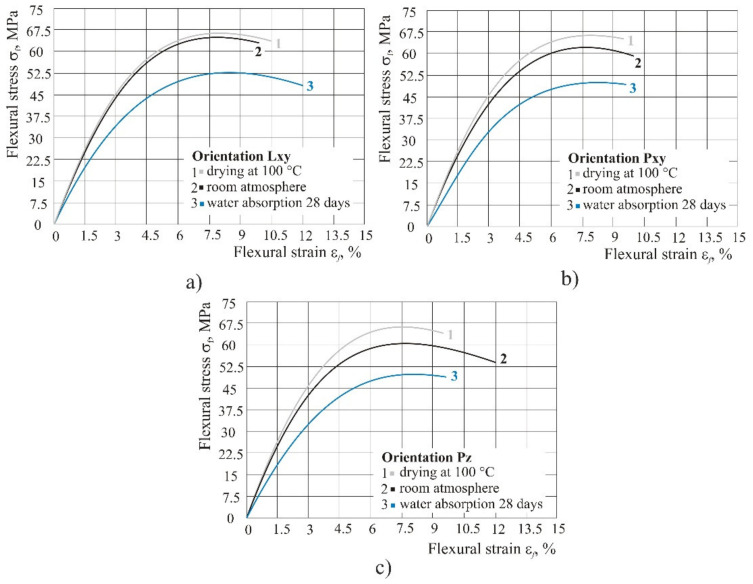
Comparison of different atmospheric conditions (dry, normal and 100% humid atmosphere): (**a**) orientation Lxy, (**b**) orientation Pxy, (**c**) orientation Pz.

**Figure 12 polymers-14-02355-f012:**
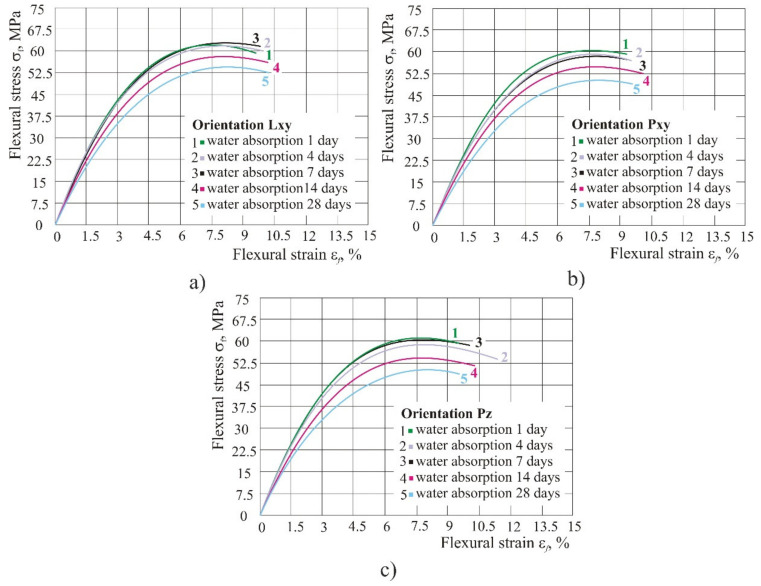
Diagram of flexural stress–strain in the case of water absorption: (**a**) orientation Lxy, (**b**) orientation Pxy, (**c**) orientation Pz.

**Figure 13 polymers-14-02355-f013:**
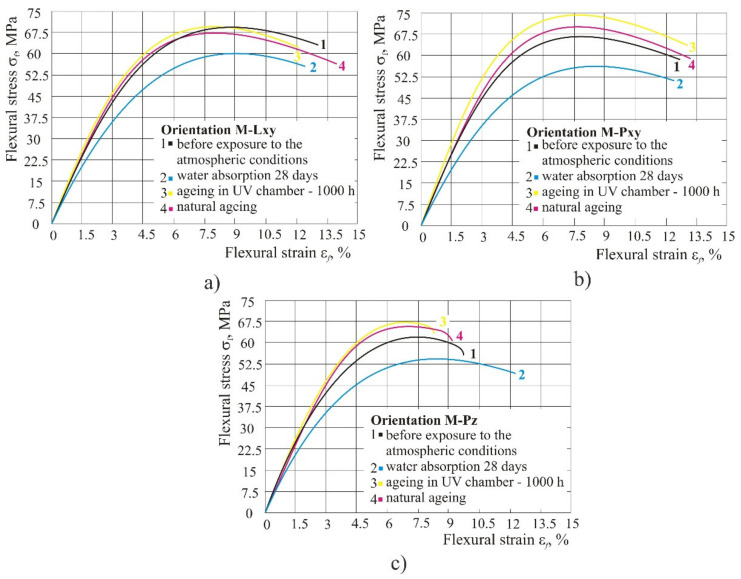
Impact of different atmospheric conditions on mixed material: (**a**) orientation Lxy, (**b**) orientation Pxy, (**c**) orientation Pz.

**Figure 14 polymers-14-02355-f014:**
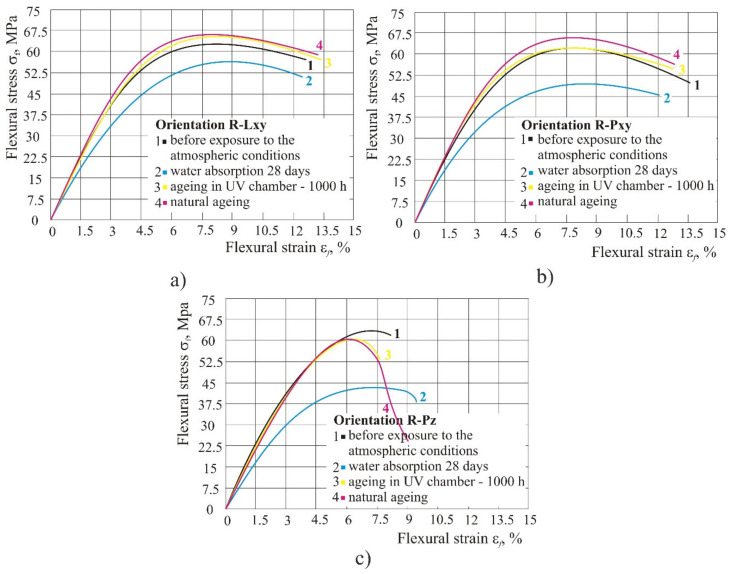
Impact of different atmospheric conditions on recycled material: (**a**) orientation Lxy, (**b**) orientation Pxy, (**c**) orientation Pz.

**Figure 15 polymers-14-02355-f015:**
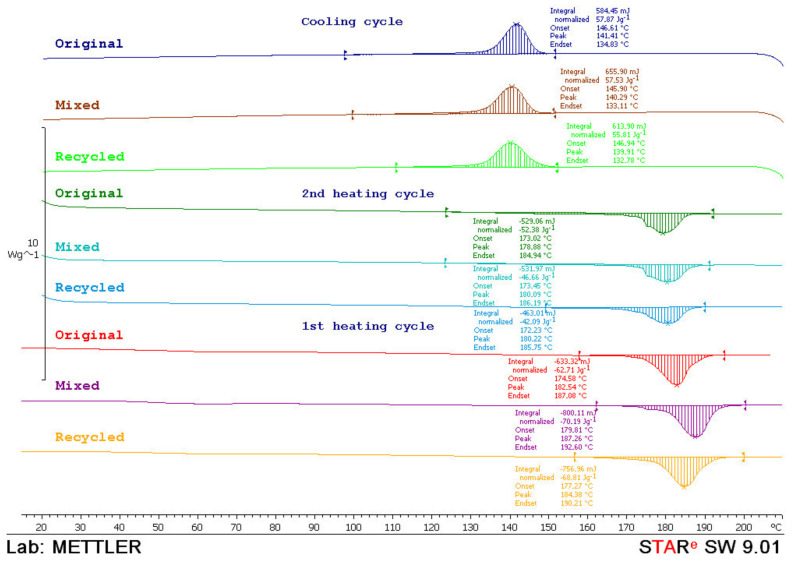
DSC thermograms of original, mixed and recycled polyamide obtained during the first and second heating cycle and the cooling cycle.

**Figure 16 polymers-14-02355-f016:**
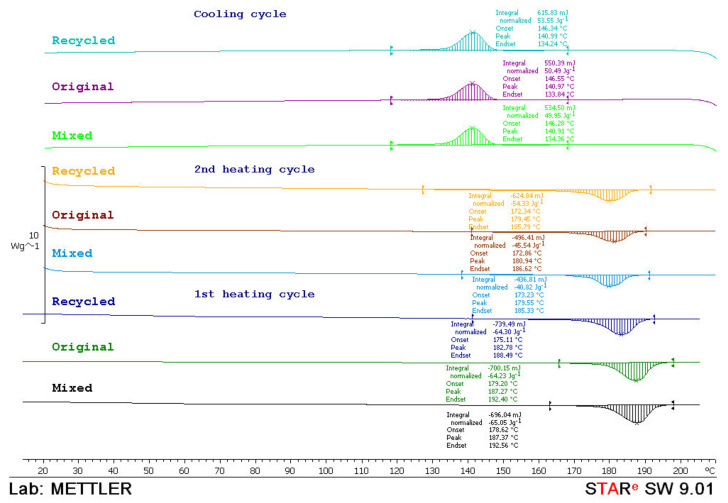
DSC thermograms of original, mixed and recycled polyamide after ageing in UV chamber 1000 h obtained during the first and second heating cycle and the cooling cycle.

**Figure 17 polymers-14-02355-f017:**
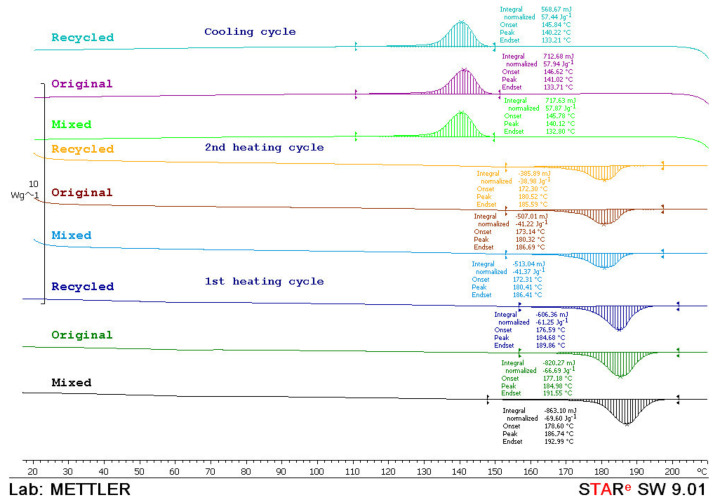
DSC thermograms of original, mixed and recycled polyamide after water absorption 28 days obtained during the first and second heating cycle and the cooling cycle.

**Figure 18 polymers-14-02355-f018:**
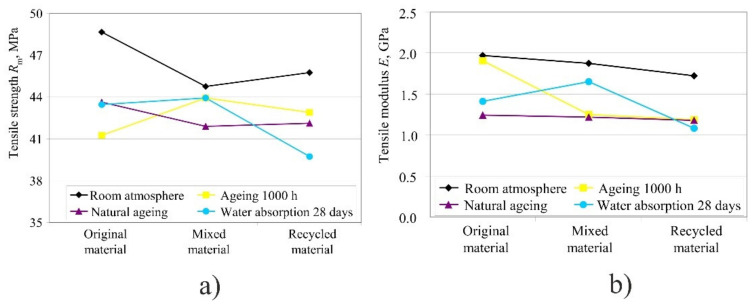
Impact of weathering conditions on tensile strength and modulus in different types of materials: (**a**) diagram for tensile strength, (**b**) diagram for tensile modulus.

**Figure 19 polymers-14-02355-f019:**
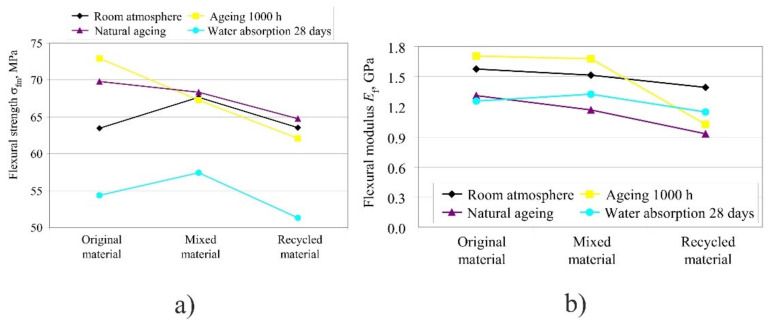
Impact of atmospheric conditions on flexural strength and modulus in different types of materials: (**a**) diagram for flexural strength, (**b**) diagram for flexural modulus.

**Table 1 polymers-14-02355-t001:** Results of DSC measurements.

	1. Heating Cycle		2. Heating Cycle		Cooling Cycle	
	*T*_m_ (°C)	*ΔH*_m_ (J/g)	*T*_m_ (°C)	*ΔH*_m_ (J/g)	*T*_c_ (°C)	*ΔH*_c_ (J/g)
	Before exposure to weathering conditions					
Original	182.54	62.71	178.88	52.38	141.41	57.87
Mixed	187.26	70.19	180.09	46.66	140.29	57.53
Recycled	184.38	68.81	180.22	42.09	139.91	55.81
	After ageing in UV chamber 1000 h					
Original	187.27	64.23	180.94	45.54	140.97	50.49
Mixed	187.37	65.05	179.55	40.82	140.91	49.95
Recycled	182.78	64.30	179.45	54.33	140.99	53.55
	After water absorption 28 days					
Original	184.98	66.69	180.32	41.22	141.02	57.94
Mixed	186.74	69.60	180.41	41.37	140.12	57.87
Recycled	184.68	61.25	180.52	38.98	140.22	57.44

## Data Availability

Not applicable.

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
