# Peer review of "Influence of Atmospheric Conditions on Mechanical Properties of Polyamide with Different Content of Recycled Material in Selective Laser Sintering"

_polymers, 2022, doi:10.3390/polym14122355_

Round 1
Reviewer 1 Report
The paper reports the influence of UV and moisture conditions on mechanical properties of PA12 parts printed with different content of recycled material in selective laser sintering. The work provides considerable insights to the topic. The paper is well structured and presented systematically. However, there are several minor modifications suggested: Please refer to the attached file.

Author Response
Dear reviewer,
Thank you for your review and effort. Every reviewer’s suggestion is valuable and leads to better quality of the paper. Answers to your review are given in PDF.

Reviewer 2 Report
Dear Authors, thank you very much for your interesting paper. It is well-written and robust. One minor comment: please use "Authors and al. " instead of "The authors of [4]". Use their name in the text
Author Response
Dear reviewer,
Thank you for your review and effort. Every reviewer’s suggestion is valuable and leads to better quality of the paper. We have changed citations according to your comment.
Reviewer 3 Report
This paper is definitely interesting for a limited number of readers, working with SLS of PA12 and more generally with 3D printing technology. This is not a significant number of readers but for people working within this research area paper is very interesting and important - gives an real and verified answer how you can practically improve your work efficiency, really apply circular economy and in simple in practice - save material and reduce costs.
Please correct lines 101; 104 to DSC (as it is in line 293). Please change in line 140 force P to power P. Remove space in lines 153; 174; 209; 297; 320; 417; 459 between digits and oV (should be in line 153 23oC). Please add some info about PA12 mixing conditions (for 50/50 ratio).
Author Response
Dear reviewer,
Thank you for your review and effort. Every reviewer’s suggestion is valuable and leads to better quality of the paper.
Please correct lines 101; 104 to DSC (as it is in line 293).
Corrected to DSC.
Please change in line 140 force P to power P.
Corrected to laser power.
Remove space in lines 153; 174; 209; 297; 320; 417; 459 between digits and oV (should be in line 153 23oC).
According to source below, °C unit is written with the space between the number and the unit:
https://www.nist.gov/pml/weights-and-measures/writing-si-metric-system-units
Please add some info about PA12 mixing conditions (for 50/50 ratio).
In text we added mixing conditions:
In the mixed material, original and recycled material in ratio 50%:50% were mixed in an industrial mixer at the room temperature. Duration of mixing process was 60 minutes.